# Business Incentives for Local Economic Development

**Maša Trinajstić** [1],*[ID], **Marinela Krstinić Nižić** [1][ID] **and Nada Denona Bogović** [2][ID]

1   Faculty of Tourism and Hospitality Management, University of Rijeka, Primorska 46,
    51410 Opatija, Croatia; marikn@fthm.hr
2   Faculty of Economics and Business, University of Rijeka, Ivana Filipovića 4, 51000 Rijeka, Croatia;
    nada.denona@ri.t-com.hr
*   Correspondence: masat@fthm.hr

**Abstract:** The main role of local development policy is to create a favorable business environment and new jobs, thus contributing to economic development. Creating a positive business environment to act as a pull factor for new businesses is of great importance, because entrepreneurship conduces to the rise in the supply of goods and to an increase in citizens' income and revenues of local budgets. This paper therefore examines the main goals of economic development in the towns and municipalities of the Republic of Croatia, as well as the tools used by local governments to encourage entrepreneurship and new businesses. Using a questionnaire, mayors and deputies of towns and municipalities were surveyed, and the sample covered 131 towns and municipalities. The research methodology included a descriptive analysis and the Kruskal–Wallis test. The results indicate that improving the quality of life of the residents, creating new jobs, and retaining the residents in the town or municipality were identified as the most important goals of economic development. To encourage entrepreneurship and new jobs, towns and municipalities most often simplify regulations, reduce local taxes, and introduce various benefits and incentives. The contribution of the paper is manifested in providing useful guidance to local governments to improve the business environment.

**Keywords:** local economic development; entrepreneurship; business environment; business incentives; residents' quality of life; Croatia

## 1. Introduction

Local economic development (LED) can be defined as a process in which partners in the public, business, and non-government sectors work together to create better conditions for economic growth and job creation (World Bank 2001). The LED concept is derived from the bottom-up management model, created in response to the traditional top-down model, which has failed to resolve many negative regional and local processes and to reduce differences in under-development. The concept of local economic development presupposes that local government, the private sector, and the local population are much more familiar with their own needs, advantages, and opportunities. This means that they can use local resources optimally, thus contributing to job creation, a positive entrepreneurial environment, and a better quality of life for the local population (Karaman Aksentijević et al. 2019). It should be emphasized that the LED process requires good communication, active participation, and cooperation between all stakeholders (Trinajstić 2021).

Favorable local business conditions are required for a company to achieve progress. Local government is the closest to people and matters of local development, thus having a key role in creating a favorable environment required for business success (Swinburn et al. 2006). In recent years, local governments have become more involved in entrepreneurship programs supported by the national government with the aim of creating local jobs (Bartik 2017). However, local government has to define development goals that represent the basis for decision making in order to ensure appropriate business conditions (Bartik 2003).

Towns and municipalities in the Republic of Croatia play a locally relevant role in directly fulfilling its citizen's needs: neighborhood and residential maintenance, spatial and urban planning, municipal management, social care, primary healthcare, primary education, and environmental protection and improvement (Law on Local and Regional Self-Government 2020). In addition to the activities prescribed by law, a part of the towns and municipalities is engaged in the Business Favorable Certification Program for towns and municipalities (CLER 2014). The program requires certain standards to be met, i.e., some sort of guarantee that existing entrepreneurs, as well as potential investors, will be provided services and information within the purview of the local government in such a way that facilitates entrepreneurship and activities. This means that entrepreneurs can expect help and a partner relationship with the town or municipality. Such initiatives are important for strengthening and supporting the local economy, i.e., local economic development. Additionally, the local government has to secure appropriate infrastructure: roads, water supply, energy supply, waste management, and information and communication technology, so these include any requirements for economic and entrepreneurial activity (UCLG 2016).

In other words, every local government wants to act in the interest of its citizens and attempts to increase the quality of life and employment, improve environmental protection, and provide high-quality public services (Cárcaba et al. 2017).

This paper explores the main goals of economic development in towns and municipalities in Croatia, as well as the tools local governments use to encourage entrepreneurship and business in order to create more jobs. The research questions thus include: What are the main goals of economic development, and is there a difference in goals regarding the size of towns and municipalities? What are the tools by which local governments may interact with private entrepreneurs to encourage entrepreneurship? Do local governments sufficiently participate in improving infrastructure and supporting the local economy?

A questionnaire was used to survey mayors and deputy mayors of 131 towns and municipalities. The paper shows that support for the development of entrepreneurship presents a key component in facilitating local economic development. The paper also aims to provide useful guidelines for local governments for improving the business conditions.

The structure of the paper is as follows: after the introduction, the second section describes the available literature. The third section describes the data and research methodology. This is followed by a presentation of the results and the discussion. The final section is the conclusion, which also includes the limitations of the research as well as guidelines for future research.

## 2. Theoretical Background

There are three important elements of local economic development: achieving local economic stability, ensuring a diverse cross-sector structure of the local economy, and promoting local sustainability (Leigh and Blakely 2017).

Achieving economic stability and creating a diverse cross-sector structure of the local economy are typically the primary elements of planning local economic development (Dissart 2003). Cross-sector diversity is assumed to increase economic stability, i.e., the presence of multiple sectors and businesses in a town reduces fluctuations in employment (Chen 2020). To achieve economic stability and create new jobs, a town must find a way to encourage new businesses opening, sustain and expand existing local businesses, attract new investments, and invest in human resources. Creating a positive entrepreneurial environment to act as a pull factor for large businesses is of great importance. Towns with only one or a small number of businesses are not safe from workforce fluctuation. Towns cannot depend on one sector alone, and they must provide better, continuous, and permanent opportunities for their inhabitants' employment. Erkuş-Öztürk and Terhorst (2018) state that towns where tourism is the dominant economic activity may diversify. Over time, the development of tourism stimulates the growth of related and unrelated industries, thus resulting in economic diversification.

Towns must place more effort on improving the employment process, service quality, and quality of life in general; improve infrastructure; and create new jobs to improve the society's sustainability (Krstinić Nižić et al. 2019). Furthermore, every town or municipality must identify opportunities and strengths and resource availability and must define the main goals for its economic development. In accordance with the defined goals, local governments enact appropriate local economic development policies (Bartik 2003; Reese and Fasenfest 1997). Figure 1 features the main goals of local development policy.

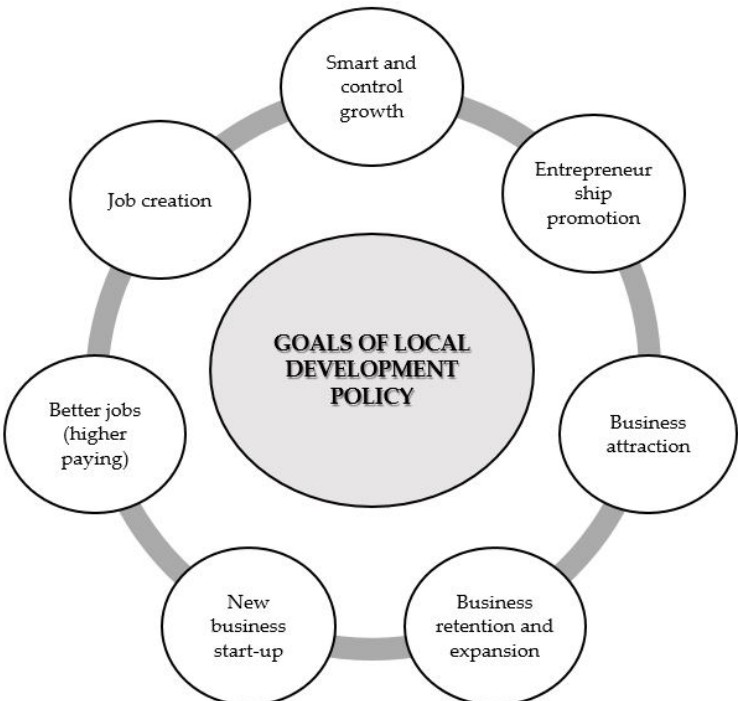

**Figure 1.** Main goals of local development policy. Source: author's adapted from (Bartik 2020; Leigh and Blakely 2017).

The main development goals have to be structured according to the needs of the local inhabitants and businesses. Further, each sector has an impact on local economic development (Leigh and Blakely 2017; Acs et al. 2016; Stenvall et al. 2022). Table 1 below shows how the public, private, and civil sectors can use their tools to improve the process of local economic development.

**Table 1.** Impact on LED by sectors.

| | Tools |
|---|---|
| **Public sector** | simplify regulations, offer tax reliefs, introduce other various fiscal benefits (e.g., removing municipal/local taxes and fees), provide financial support, invest in entrepreneurial infrastructure (zones, technology parks, incubation centers) |
| **Private sector** | encourage private/public partnership, actively participate in decision-making and social dialogue, encourage and support invention and innovation, share its business knowledge and expertise |
| **Civil society** | raise awareness of social issues, empower local communities to develop new programs to meet their own needs, increase accountability, promote participation in decision-making, directly engage in service delivery |

Source: author's analysis based on literature.

Local economic development is the synergy effect of many processes initiated and implemented by various actors. Local government plays the main role in stimulating local development, and it strives to achieve the best possible living conditions in cooperation with the private sector and residents.

When analyzing economic structure and defining economic development priorities, local governments have to consider the following (Leigh and Blakely 2017):

- Define which sectors play a key role in the local economy when it comes to jobs, sales, taxes, and connections with other industries like agriculture, forestry, and crafts.
- Identify the links between the local and external economy to define how agile the local sectors and infrastructure are in reacting to changes in the regional, national, and international economy.
- Evaluate local potential that may ensure economic growth and stability.
- Research unpredictable situations that may have a significant impact on jobs, public income and spending, economic productivity, work quality, as well as the quality of life.

As mentioned above, in order to achieve its priorities in economic development, towns and municipalities must provide a favorable business environment. Many empirical studies support the existence of a positive relationship between entrepreneurship and economic development (Chen 2014; Hessels and van Stel 2011; Lee and Xin 2015). Entrepreneurship is also highly promoted in developing countries as a strategy for achieving economic development (UNCTAD 2015), against the background of high levels of poverty and unemployment. The effects of entrepreneurship on economic development can vary. Van Stel et al. (2005) and Zaki and Rashid (2016) found that entrepreneurship has a positive relationship with economic growth in developed countries but a negative relationship in developing countries. Furthermore, Munemo (2012) concluded that most of the enterprises created in developed countries would create more jobs than those created in developing countries. Mwatsika (2021) researched the knowledge on entrepreneurship and entrepreneurial activities in Malawi, a developing country in Southern Africa. The analysis involved 337 businesses and revealed that entrepreneurship is mostly defined as the starting and running of one's own business, self-employment, and creating new jobs. Small economic activities creating income are dominant, along with micro and small businesses. Entrepreneurship is on the rise in Croatia, primarily when it comes to small enterprises (Rajsman and Petričević 2013), but it still falls behind the economic efficiency of such enterprises in the most developed countries in the EU and the world. The benefit of entrepreneurship on employment and total economic development is evident, and as such, Croatia, just like other countries and especially less developed ones, has to upkeep the measures for improving the entrepreneurial environment and developing entrepreneurship in general.

The role of local governments in facilitating entrepreneurship is also very significant. Local governments want to promote entrepreneurship to facilitate local development and prosperity (Isenberg 2010; Banaszewska et al. 2022). It can be said that local governments have become responsible for stimulating the dynamic growth of local enterprises as a result of the decentralization processes taking place in many European countries (Skica et al. 2013). In other words, governments are important institutional players influencing entrepreneurial activities (Minniti 2008; Zahra and Wright 2011; Xing et al. 2018). Wołowiec and Skica (2013) examined the instruments used by local governments to support entrepreneurship in municipalities in Poland. The authors concluded that the instruments vary based on the size of the municipality and that financial instruments, like tax breaks, are less important to entrepreneurs than legal and technical instruments, such as better infrastructure. Stojčić et al. (2022) analyzed the economic effects of entrepreneurial zones in towns and concluded that such zones bring positive results in host towns and neighboring towns, indicating that their effect is local by nature.

In Croatia, Jurlina Alibegović et al. (2019) explored the impact of entrepreneurial zones on local economic outcomes. The research results revealed that entrepreneurial zones are

important for local economic outcomes. The results of their empirical analysis confirmed that units of local self-government with entrepreneurial zones are more successful than those that do not have such a zone.

Upon a review of existing literature, the positive effect of entrepreneurship on local development is evident. The purpose of this paper is thus to determine the extent to which local governments in Croatia support local economy, i.e., the tools and methods used to create a favorable business environment and encourage entrepreneurship in towns and municipalities. Furthermore, the paper also presents the main development goals of towns and municipalities. The goals may vary based on the size of the town or municipality, and the goals serve as the basis for local governments to select the appropriate tools to encourage a business environment.

The authors have set two hypotheses:

**Hypothesis 1 (H1).** *There are statistically significant differences in town and municipality development goals based on size of the town or municipality.*

**Hypothesis 2 (H2).** *Local governments participate in supporting the local economy and infrastructure improvement to a large degree.*

### 3. Data and Methodology

As a member of the European Union, Croatia is now divided into four statistical NUTS 2 regions: Pannonian, Adriatic and Northern Croatia, and the city of Zagreb. This categorization has been in effect since January 2021. Previously, Croatia was divided into two statistical regions: Continental and Adriatic Croatia. As the research outlined in this paper was conducted in 2019 and 2020, two regions were used in the analysis. Based on the same classification, the Republic of Croatia (country) is on the NUTS 1 level, and counties are on the NUTS 3 level.

The NUTS 2 level is a territorial criterion through which the research aimed to encompass the entire territory of Croatia. Population size is also one of the criteria, so the participating towns and municipalities were split into three groups: less than 5000 residents, 5000–15,000 residents, and over 15,000 residents. According to the Ministry of Justice and Public Administration (2021), Croatia comprises 127 towns, 428 municipalities, and the City of Zagreb, which bears the special status of both city and county, totaling 556 towns and municipalities. Using stratified sampling, a random sample of 30% of the full set was selected. The questionnaire was thus sent to 167 mayor and deputy mayor email addresses. At the end of the polling period, 131 valid responses were acquired, amounting to 78%.

The research was conducted between October 2019 and May 2020. The data were gathered through an online survey sent by email to mayors and deputy mayors of selected towns and municipalities. The respondents were asked to define the key goals of economic development, which tools they use to encourage new jobs and entrepreneurship, and which tools to retain and expand existing businesses. The questions were formed in the form of multiple-choice, where respondents could mark up to three answers. Then, for the next two questions, a 7-degree Likert scale (1: exceptionally low, 7: exceptionally high) was used to determine the extent to which towns and municipalities participate in supporting the local economy and infrastructure improvement. The question related to strengthening the local economy included 6 items, and the question related to infrastructure improvement included 5 items.

The acquired data were analyzed in the Stata 14.2 software(StataCorp, Texas, TX, USA). A descriptive analysis and bivariate statistical methods were used. To check whether the distribution of towns and municipalities according to the region and population size obtained in the sample was significantly different from that in the population, the chi-square test was used. The results showed that there was no statistically significant difference for the region ($p = 0.2273$) and for population size ($p = 0.0808$) between the observed and expected distribution and thus confirmed the correctness of the sample selection. The

Skewness and Kurtosis test was used to examine data normality. As the test did not show the normal distribution for a larger number of variables, nonparametric methods were used to conduct the analysis. Statistical significance of the differences in economic development goals based on town and municipality size was determined using the Kruskal-Wallis test. In cases where the results showed statistically significant differences, the *Post hoc* test by Dunn was used to determine which towns and municipalities showed differences. Descriptive methods were used to determine the participation of local government in supporting the local economy and improving infrastructure.

## 4. Results and Discussion

### 4.1. Descriptive Statistics

There are a total of 336 towns and municipalities in Continental Croatia, and there are a total of 220 towns and municipalities in Adriatic Croatia. The following Tables 2 and 3 show the profile of respondents by NUTS 2 level (2020) and by population.

**Table 2.** Survey respondents by NUTS 2 level.

| Respondents by NUTS 2 Region | N | % |
|:---:|:---:|:---:|
| Continental Croatia | 71 | 54.20 |
| Adriatic Croatia | 60 | 45.80 |
| **Total** | **131** | **100.00** |

Source: authors' calculation.

**Table 3.** Survey respondents by population size.

| Respondents by Population Size | N | % |
|:---:|:---:|:---:|
| Less than 5000 residents | 80 | 61.06 |
| 5000–15,000 residents | 41 | 31.30 |
| Over 15,000 residents | 10 | 7.64 |
| **Total** | **131** | **100.00** |

Source: authors' calculation.

Towns and municipalities in Continental and Adriatic Croatia are similarly represented, i.e., in Continental Croatia, 71 questionnaires were filled in correctly, while in Adriatic Croatia, the number was 60.

The results show that most respondents were mayors and deputy mayors of small towns and municipalities (61.06%), and the fewest respondents stemmed from large towns and municipalities (7.64%). While this percentage may appear low, it should be emphasized that Croatia only has three towns with over 100,000 residents and a total of 37 towns with over 15,000 residents. This means that 27% of large towns filled out the questionnaire.

### 4.2. Kruskal–Wallis and Post Hoc Test

Towns and municipalities must choose the best tools and strategies to attract jobs and create a favorable business environment in line with economic development goals (Morgan 2010). The respondents in this research were asked to select their main economic development goals, which are defined in strategic development plans that are usually adopted for a period of 5–10 years. The question encompassed 12 goals (author's processing based on Morgan 2009): Create new jobs; Better paid jobs; Higher taxes; Keep and expand existing businesses; Encourage opening new businesses; Attract retail and services currently not available at the location; Promote small enterprises and crafts; Smart growth; Promote social and economic equality; Retain residents; Improve residents' quality of life; Create wealth.

Figure 2 shows the main economic development goals in selected towns and municipalities.

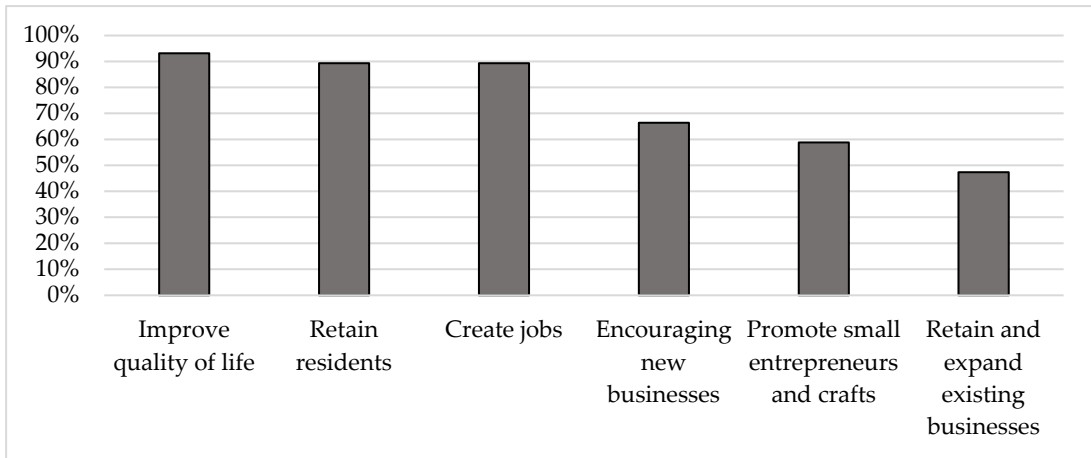

**Figure 2.** Main economic development goals in selected towns and municipalities Source: authors' analysis.

The participants selected improving quality of life (93.13%), creating new jobs (89.31%), and retaining residents (89.31%) as the three main economic development goals. According to Bartik (2020), local job creation yields significantly higher employment rates. The fourth goal was to encourage the opening of new businesses (66.41%), followed by promoting small enterprises and crafts (58.78%). In addition to facilitating new businesses, towns and municipalities also aim to retain and expand existing businesses (47.33%). It is interesting to see that retaining residents is one of the key goals. According to the most recent data by the Croatian Bureau of Statistics (2021), numerous Croatian towns and municipalities are facing population loss or drain. The negative demographic trends also cause negative economic effects, so achieving this goal is paramount.

Table 4 shows the differences in economic development goals based on the town or municipality size.

**Table 4.** Results of the *Kruskal–Wallis* test (differences in economic development goals based on population size).

| Goals | df | F | *p*-Value |
|---|---|---|---|
| Job creation | 2 | 9.917 | 0.0070 * |
| Better paid jobs | 2 | 22.733 | 0.0001 * |
| Keep and expand existing businesses | 2 | 10.178 | 0.0062 * |
| Encouraging new businesses | 2 | 12.085 | 0.0024 * |
| Smart growth | 2 | 6.905 | 0.0317 ** |

Note: * significant at the 1% level, ** significant at the 5% level. Source: authors' calculation.

The results showed statistically significant differences in development goals between towns and municipalities based on their size. The test encompassed all 12 goals, but the table only displays the results indicating statistically significant differences.

Figure 3 shows that there was a statistically significant difference between small, medium, and large towns concerning goal 2, "better paid jobs" ($p < 0.01$). Large towns placed emphasis on securing new jobs and better paid jobs for their residents. A statistically significant difference could also be seen between small and large towns on goal 5, "facilitating the founding of new businesses" ($p < 0.05$). Large towns invest more resources in encouraging new businesses through startup incubators, collaboration with the developmental agency, and various other subsidies, while small towns are more focused on retaining existing businesses. A statistically significant difference has also been noted between small and large towns on goal 8, "smart growth" ($p < 0.05$). Larger towns aim more to improve public services by using smart technologies and solutions.

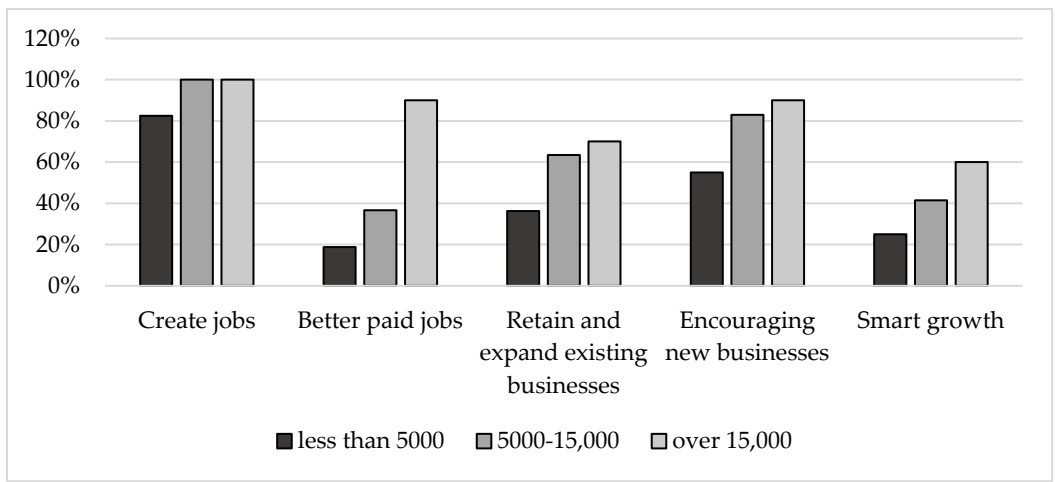

**Figure 3.** Results of Post-hoc test. Source: authors' calculation.

*4.3. Supporting Local Economy and Infrastructure*

　　Towns and municipalities need to act to strengthen their economy by creating a positive and motivating business environment, i.e., they have to support the local economy. The analysis showed that the most commonly used methods and tools are simplifying regulation, reducing taxes, and implementing various benefits and incentives (e.g., no fees for municipal contributions, smaller or no local tax during the first year of business). Additionally, local governments must encourage the opening of new businesses; support small and medium businesses; attract foreign investments by securing appropriate infrastructure, services, and access to resources; as well as encourage public–private partnerships as one of the means to solve economic development issues.

　　The following Table 5 shows the extent of town/municipality participation in supporting and strengthening local economy.

**Table 5.** Participation of local government in supporting the local economy.

| Support for Local Economy | N | Mean | Std. Dev. |
|---|---|---|---|
| Creating a positive business environment (lower taxes, various benefits, simpler regulations) | 131 | 4.72 | 1.636 |
| Supporting small and medium businesses (entrepreneurial zones and centers) | 131 | 4.70 | 1.656 |
| Encourage the opening of new businesses | 131 | 4.56 | 1.739 |
| Attracting foreign investments (domestic and international) | 131 | 4.13 | 1.867 |
| Creating/supporting public–private partnerships | 131 | 3.65 | 1.902 |
| Tourism development | 131 | 5.08 | 1.639 |
| **Total mean score** | | **4.47** | - |

Source: authors' calculation.

　　The analysis shows that the total mean score was 4.47, indicating that local government participation in supporting and strengthening local economy is high. The highest mean score was recorded on the "Tourism development" item, followed by "Creating a positive business environment" and "Supporting small and medium businesses". The lowest score recorded was on the "Creating/supporting public–private partnerships" item. It is interesting to note that many towns and municipalities desire to strengthen their economy by developing tourism. This confirms the fact that Croatia is a tourism-oriented country and that tourism is the leading economic activity. Over half of the towns and municipalities considered small and medium entrepreneurship to be what drives job creation, that communication with local entrepreneurs should be active, and that they should engage in joint activities. The high score stems from this. Even though public–private partnership had the lowest mean score and is still not sufficiently represented in Croatia, there are indications

that towns and municipalities recognize the potential of such partnerships. By combining the resources of the public and private sector through public–private partnerships, common interests for both can be achieved for the purpose of developing a town or municipality.

Appropriate infrastructure is required to develop and strengthen the economy and to achieve business activity growth. Infrastructure has always been one of the top priorities of local governments and has, in recent times, been recognized as key for economic development (Pike et al. 2017). The following Table 6 presents local government participation in developing and improving infrastructure.

**Table 6.** Participation of local government in enhancing infrastructure.

| Infrastructure | N | Mean | Std. Dev. |
|---|---|---|---|
| Construction, improvement, and expansion of entrepreneurial zones | 131 | 5.04 | 1.797 |
| Construction and improvement of traffic infrastructure | 131 | 5.63 | 1.409 |
| Improvement of the local sewer system | 131 | 5.35 | 1.822 |
| Improvement of the local power distribution system | 131 | 5.08 | 1.630 |
| Improvement of the local water distribution system | 131 | 5.63 | 1.441 |
| **Total mean score** | | **5.35** | - |

Source: authors' calculation.

The results show that the total mean score was 5.35, indicating that local government participation in enhancing infrastructure is very high. The highest mean score was recorded on the "Construction and improvement of traffic infrastructure" item, followed by "Improvement of the local water distribution system", while the lowest score was recorded on "Construction, improvement, and expansion of entrepreneurial zones". Local governments are aware of the importance of good infrastructure for both economic and social development. Improving infrastructure enables greater economic development and manufacturing diversity, and it makes the town or municipality more attractive for founding new businesses or attracting existing businesses. It also contributes to higher property values and improving overall quality of life (Puljiz 2005).

To encourage entrepreneurship and job creation, each local government has to use the methods and tools appropriate for their circumstances (position, traffic options, and resource availability). Looking at the research results, it can be concluded that local governments in Croatia create new jobs by collaborating with the regional or local development agency (64.12.%), have calls to action (60.3%), provide financial support to new businesses (42%), and conduct marketing and advertising activities (35.11%). Another measure involves reducing municipal fees for all new investments in entrepreneurial zones. In order to retain and expand existing businesses, local governments are investing in entrepreneurial infrastructure (expanding entrepreneurial zones) and are reducing or even altogether dropping municipal fees for businesses expanding their existing manufacturing facilities. To help with creating and developing small and medium businesses, towns and municipalities provide subsidies (54%), found startup incubators (22.14%), and provide assistance with marketing (23.66%). This was also confirmed in the study on a sample of Polish cities, where the authors pointed out that city authorities have a positive role in creating start-ups (Jonek-Kowalska and Wolniak 2021). Additional measures include subsidizing interest rates on credits and founding entrepreneurial centers. To attract new investors, some local governments purchase real estate and office buildings from bankrupt companies and prepare them for new investors.

## 5. Conclusions, Recommendations, and Limitations

Supporting the growth of entrepreneurship is paramount for facilitating local economic development. The strength of local entrepreneurship defines income size and type, while specific programs and measures planned and shaped by local politics and budgets can significantly contribute to the development of entrepreneurship.

This research defined the key goals of economic development for towns and municipalities in Croatia, and it confirmed that there are statistically significant differences in prioritizing goals based on town or municipality size. This confirms the first research hypothesis. All towns and municipalities listed quality of life improvement and population retention as the two primary goals. Large towns then prioritized securing better paid jobs for their inhabitants and smart growth, while small towns and municipalities prioritized promoting small businesses and crafts and retaining existing businesses.

It was further determined that local governments have a strong presence in strengthening and supporting the local economy by simplifying regulations, reducing taxes, and introducing various fiscal benefits, such as relaxing or removing municipal fees overall or in the first year of business or reducing or removing local taxes in the first year of business.

Additionally, it was determined that local governments strongly participate in improving infrastructure in accordance with local needs and that they understand the importance of strong infrastructure. This confirms the second research hypothesis.

The results of this research have confirmed some previous research in terms of the contribution of local government to supporting entrepreneurship (De Matteis et al. 2022; Jonek-Kowalska and Wolniak 2021; Skica et al. 2013). Successful private companies create prosperity in towns and municipalities. However, for a private company to make progress, favorable local business conditions are necessary. Thus, the local government plays a very important role in supporting entrepreneurship and creating the enabling environment necessary for business success. Local governments have to be reliable partners to entrepreneurs by ensuring the highest standard of services, which is very important for investors. That is the only way for towns and municipalities to be recognized by existing businesses and potential investors as favorable destinations for new projects (CLER 2014). New investments are required to increase employment, and new jobs are paramount for improving the quality of life of residents, which was the highest scoring goal for economic development.

In order to continue supporting the local economy and develop a favorable business environment, further recommendations are that towns and municipalities in Croatia provide entrepreneurs with assistance in obtaining licenses and permits, adjust legal and sublegal regulations to attract new investors, implement stimulative measures, such as reducing fees or lowering rent for entrepreneurs, organize or co-organize (in partnership with educational institutions) training and workshops catering to the needs of entrepreneurs, enact appropriate spatial planning, and ultimately bring business culture to the next level.

This paper's contribution lies in the analysis of the key goals for economic development in towns and municipalities, as well as the tools and methods used by local governments to encourage entrepreneurship and develop a favorable business environment. The research also shows that priorities for facilitating entrepreneurship set by local governments match the main goals of economic development in towns and municipalities. For future scientific endeavors, this paper constitutes a contribution in the theoretical sense, provides a basis for expanding on the subject, and provides local governments with useful results for improving the local business environment.

Finally, it is necessary to point out the limitations of this research. The research was conducted by surveying mayors and deputy mayors. The results reflect their personal attitudes on the subject matter. The recommendation for further research pertains to research methodology. Using available secondary data would enable comparisons not just on the level of towns and municipalities in Croatia, but also to compare cities across the European Union with cities and towns in Croatia. An additional recommendation is that the survey also involves entrepreneurs active in target towns and municipalities. The research could also be expanded by involving other stakeholders, such as educational institutions and associations.

**Author Contributions:** Conceptualization, M.T. and M.K.N.; methodology, M.T; validation, M.T., M.K.N. and N.D.B.; formal analysis, M.T. and M.K.N.; investigation, M.T.; data curation, M.T.; writing—original draft preparation, M.T. and M.K.N.; writing—review and editing, M.K.N. and

N.D.B.; visualization, M.T. and M.K.N.; supervision, M.K.N. and N.D.B. All authors have read and agreed to the published version of the manuscript.

**Funding:** This paper has been fully supported by the University of Rijeka under the projects "Sustainable cities as carriers of economic development" (uniri-drustv-18-212), "Smart cities in function of development of national economy" (uniri-drustv-18-255-1424) and "Impact of Intangible Capital in Croatian Economy" (uniri-drustv-18-166)".

**Institutional Review Board Statement:** Not applicable.

**Informed Consent Statement:** Not applicable.

**Data Availability Statement:** Data available on request from the corresponding author.

**Conflicts of Interest:** The authors declare no conflict of interest.

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
