# Peer review of "Business Incentives for Local Economic Development"

_economies, doi:10.3390/economies10060135_

Round 1
Reviewer 1 Report
The article is easyreadable, but some remarks for corrections are:
There could be more new literature based on topic (for example in Figure 1. Main goals of economic development Source: author's processing based on Bartik 2003 104 and Morgan 2009-it is too old);
May there can appear some table with impact (with tools and methods) by each sector on LED-what exist now?
It is recommended to give more data about questionnaire - what types of questions, what their goals etc. What types of answers were used etc.
The subchapter 4.1. Descripitve stastistics - is too narrow, it requires more comments.
How does the findings of research reflect previous recent research?
Author Response
Response to Reviewer 1 Comments
Dear reviewer 1,
Thank you very much for your constructive suggestions. They are of great significance to our paper writing and scientific research. We have revised the manuscript in accordance with your concerns and comments. Our responses to your comments are summarized below. Please feel free to let us know if you have further comments or questions. Thanks a lot!
Point 1: There could be more new literature based on topic (for example in Figure 1. Main goals of economic development Source: author's processing based on Bartik 2003 104 and Morgan 2009-it is too old);
Response 1: Thank you for your comment. We added newer literature/sources.
Bartik, Timothy J. 2020. Bringing Jobs To People: Improving Local Economic Development Policies. Policy Paper No. 2020-023. Kalamazoo, MI: W.E. Upjohn Institute for Employment Research. https://doi.org/10.17848/pol2020-023.
Point 2: May there can appear some table with impact (with tools and methods) by each sector on LED-what exist now?
Response 2:
Table 1. Impacts on LED by sectors
|
|
Tools |
|
Public sector |
simplify regulations, offer tax reliefs, introduce other various fiscal benefits (e.g. removing municipal/local taxes and fees), provide financial support, invest in entrepreneurial infrastructure (zones, technology parks, incubation centers) |
|
Private sector |
encourage private/public partnership, active participate in decision-making and social dialogue, encourage and support invention and innovation, share its business knowledge and expertis |
|
Civil sector |
raise awareness of social issues, empower local communities to develop new programs to meet their own needs, increase accountability, promote participation in decision-making, directly engage in service delivery |
Source: author’s analysis based on literature
Local economic development is the synergy effect of many processes initiated and implemented by various actors. Local government plays the main role in stimulating local development and it strives to achieve the best possible living conditions in cooperation with the private sector and residents.
Point 3: It is recommended to give more data about questionnaire - what types of questions, what their goals etc. What types of answers were used etc.
Response 3: Thank you. We added more details about the questionnaire.
The respondents were asked to define the key goals of economic development, which tools they use to encourage new jobs and entrepreneurship, and which to retain and expand existing businesses. The questions were formed in the form of multiple-choice where respondents could mark up to three answers. Then, for the next two questions, a 7-degree Likert scale (1 – exceptionally low, 7 – exceptionally high) was used to determine the extent to which towns and municipalities participate in supporting the local economy and infrastructure improvement. The question related to strengthening the local economy included 6 items, and the question related to infrastructure improvement, 5 items.
Point 4: The subchapter 4.1. Descriptive statistics - is too narrow, it requires more comments.
Response 4: Thank you, we added more coments.
There are a total of 336 towns and municipalities in Continental Croatia, and a total of 220 towns and municipalities in Adriatic Croatia.
Towns and municipalities in Continental and Adriatic Croatia are similarly represented, i.e. in Continental Croatia, 71 questionnaires were filled in correctly, while in Adriatic Croatia the number is 60.
Point 5: How does the findings of research reflect previous recent research?
Response 5: The results of this research have confirmed some previous research in terms of the contribution of local government to supporting entrepreneurship (de Matteis et al., 2021, Jonek-Kowalska and Wolniak, 2021, Tomasz et al., 2013). Successful private companies create prosperity in towns and municipalities. However, for a private company to make progress, favorable local business conditions are necessary. So, the local governments play a very important role in supporting entrepreneurship and creating the enabling environment necessary for business success.
Reviewer 2 Report
The research problem discussed in the article is topical and crucial both from the theoretical and practical points of view. Though the main task of local government is to meet the collective needs of the residents, in many cases, the catalogue of implemented activities is much broader. Within the scope of their competencies, local authorities influence the structure, pace and directions of evolution of the economic entities located in the commune, thus contributing to economic development. The expansion of entrepreneurship is a crucial determinant of local growth because it conduces to the rise in the supply of goods and jobs. It also contributes to an increase in the citizens' income and revenues of local budgets, thanks to which it supports the municipal investment policy and enabling better satisfaction of the needs of the local community. Therefore each study about creating favourable location conditions for the development of enterprises helps us to enrich our knowledge on this subject, which should be positively noted and emphasized. However, I would like to draw your attention to a few issues that require improvement.
In my opinion, the title does not fully reflect the content covered in the article and therefore requires rewording. In the content, the authors focus primarily on the goals of local development policy, tools for strengthening the local economy, and local government participation in developing and improving infrastructure. However, there is little information on specific methods and tools used by local governments to increase the scale of entrepreneurship (and this is what I expected from the title 'Business incentives for local economic development: methods and tools used by local governments').
I also have reservations about the phrase 'local development goals' that appears in the text, among others in the first sentence of the summary, the purpose, and the research question. The statement: 'One of the main goals of local economic development is to create a favored business environment and new jobs while preserving the quality of life of residents' is, in my opinion, incorrect. Local economic development is the synergy effect of many processes initiated and implemented by various actors (both organizations and individuals). It is the main goal of every self-government community that strives to achieve the best possible living conditions through the effective management of endogenous resources and exogenous factors. Local government policy plays the main role in stimulating local development. In such an approach creating a favored business and social environment is the main goal of local development policy (not of local development). The lack of distinction between local development and local development policy goals is particularly evident in Figure 1. The main goal of local development is to improve the quality of inhabitants' life and the competitiveness of the territorial unit. New jobs or the promotion of entrepreneurship are the goals of the local development policy. Their achievement results in local development. Confirmation of this we can find in the text, e.g. 'Entrepreneurship is also highly promoted in developing countries as a strategy for achieving economic development (UNCTAD 2015)' (lines 124-126) or 'Supporting the growth of entrepreneurship is paramount for facilitating local economic development'(line 328-329). Probably the statement is a mental abbreviation and thus requires clarification.
Another mental shortcut appears in the sentence: 'The respondents in this research were asked to select their main long-term economic development goals' (lines 225 and 226). We must remember that the respondents represent the local community and manage public affairs on its behalf. On the other hand, they are private persons and 'ordinary' residents of towns/municipalities. The expression used in the article ('their main long-term economic development goals') suggests that the respondents answered not as officials but as inhabitants, but this was not the case. The questions in the survey questionnaire concerned the long-term goals of the local development policy, not their individual goals. So this also needs improvement.
For full methodological correctness, it would be worth supplementing the study by checking (using the chi-square test) whether the distribution of local governments according to population size obtained in the sample was significantly different from that in the population. A statistically insignificant result will indicate no significant differences between the observed and the expected distribution and will confirm the correctness of the sample selection.
There is a lack of discussion and reference to conclusions formulated by other researchers in section 4. Results and discussion, especially that the subject is quite popular.
It would also be worth paying attention to the effects of the activities (effectiveness of the policy). However, I am aware that such an extension of the scope is impossible due to text length requirements.
Author Response
Response to Reviewer 2 Comments
Dear reviewer 2,
Thank you very much for your constructive suggestions. They are of great significance to our paper writing and scientific research. We have revised this manuscript in accordance with your concerns and comments. Our responses to your comments are summarized below. Please feel free to let us know if you have further comments or questions. Thanks a lot!
Point 1: In my opinion, the title does not fully reflect the content covered in the article and therefore requires rewording. In the content, the authors focus primarily on the goals of local development policy, tools for strengthening the local economy, and local government participation in developing and improving infrastructure. However, there is little information on specific methods and tools used by local governments to increase the scale of entrepreneurship (and this is what I expected from the title 'Business incentives for local economic development: methods and tools used by local governments').
Response 1: Thank you very much for your comment. We agree, further is new title:
Business incentives for local economic development
Point 2: I also have reservations about the phrase 'local development goals' that appears in the text, among others in the first sentence of the summary, the purpose, and the research question. The statement: 'One of the main goals of local economic development is to create a favored business environment and new jobs while preserving the quality of life of residents' is, in my opinion, incorrect. Local economic development is the synergy effect of many processes initiated and implemented by various actors (both organizations and individuals). It is the main goal of every self-government community that strives to achieve the best possible living conditions through the effective management of endogenous resources and exogenous factors. Local government policy plays the main role in stimulating local development. In such an approach creating a favored business and social environment is the main goal of local development policy (not of local development). The lack of distinction between local development and local development policy goals is particularly evident in Figure 1. The main goal of local development is to improve the quality of inhabitants' life and the competitiveness of the territorial unit. New jobs or the promotion of entrepreneurship are the goals of the local development policy. Their achievement results in local development. Confirmation of this we can find in the text, e.g. 'Entrepreneurship is also highly promoted in developing countries as a strategy for achieving economic development (UNCTAD 2015)' (lines 124-126) or 'Supporting the growth of entrepreneurship is paramount for facilitating local economic development'(line 328-329). Probably the statement is a mental abbreviation and thus requires clarification.
Response 2: Thank you, we agree. During the translation, the phrases were clumsily derived.
The main role of local development policy is to create a favorable business environment and new jobs, thus contributing to economic development. Creating a positive business environment to act as a pull factor for new businesses is of great importance because entrepreneurship conduces to the rise in the supply of goods, and to an increase in citizens’ income and revenues of local budgets. This paper, therefore, examines the main goals of local development policy in the towns and municipalities of the Republic of Croatia, as well as the tools used by local governments to encourage entrepreneurship and new businesses. Using a questionnaire, mayors and deputies of towns and municipalities were surveyed, and the sample covered 131 towns and municipalities. The research methodology included descriptive analysis and the Kruskal-Wallis test. The results indicate that improving the quality of life of the residents, creating new jobs, and retaining the residents in the town or municipality were identified as the most important goals of economic development. To encourage entrepreneurship and new jobs, towns and municipalities most often simplify regulations, reduce local taxes, and introduce various benefits and incentives. The contribution of the paper is manifested in providing useful guidance to local governments to improve the business environment.
Point 3: Another mental shortcut appears in the sentence: 'The respondents in this research were asked to select their main long-term economic development goals' (lines 225 and 226). We must remember that the respondents represent the local community and manage public affairs on its behalf. On the other hand, they are private persons and 'ordinary' residents of towns/municipalities. The expression used in the article ('their main long-term economic development goals') suggests that the respondents answered not as officials but as inhabitants, but this was not the case. The questions in the survey questionnaire concerned the long-term goals of the local development policy, not their individual goals. So this also needs improvement.
Response 3: Thank you. The authors discuss the period of 4-10 years. According to the Law on Local Self-Government, mayors and deputies can manage the town or municipality in one term of 4 years. Also, on the other hand, strategic development plans of towns/municipalities are usually adopted for a period of 5-10 years, and contain economic development goals. Therefore, the authors thought of the period 5-10 years, and perhaps it was a little clumsy performed in the sentence.
The respondents in this research were asked to select their main economic development goals which are defined in strategic development plans that are usually adopted for a period of 5-10 years.
Point 4: For full methodological correctness, it would be worth supplementing the study by checking (using the chi-square test) whether the distribution of local governments according to population size obtained in the sample was significantly different from that in the population. A statistically insignificant result will indicate no significant differences between the observed and the expected distribution and will confirm the correctness of the sample selection.
Response 4: Thank you for your suggestion. We supplemented the methodology using the chi-square test.
To check whether the distribution of towns and municipalities according to the region and population size obtained in the sample was significantly different from that in the population, the chi-square test was used. The results showed that there was no statistically significant difference for the region (p=0.2273) and for population size (p=0.0808) between the observed and expected distribution and thus confirmed the correctness of the sample selection.
Point 5: There is a lack of discussion and reference to conclusions formulated by other researchers in section 4. Results and discussion, especially that the subject is quite popular.
Response 5: Thank you, we added more research.